

# Efficiency improvement of Juniper trees mass propagation through *in vitro* shoot multiplication

Alae Ahmed Jabbour[1] and Abdulaziz Alzahrani[2]

[1] Department of Biology, Faculty of Applied Science, Umm Al-Qura University, Makkah, Saudi Arabia
[2] Department of Biology, Faculty of Science, Al-Baha University, Alaqiq, Al-Baha, Saudi Arabia

## ABSTRACT

*Juniperus procera* is an endangered medicinal tree found in Saudi Arabia. Juniper trees face numerous challenges with seed production, germination and limited clonal propagation potential. Therefore, alternate techniques for reproducing *Juniperus procera* are essential for large-scale production. The main propose of the current research was establishment of an *in vitro* shoot multiplication protocol for *J. procera*. Explants were initially cultured in Murashige and Skoog (MS) media with varying combinations of benzyl amino purine (BAP), than the sprouted shoots were sub-cultured on MS media with different combination of BAP and naphthaleneacetic acid (NAA); rooting potential was examined on both MS and olive medium (OM) media supplemented with indole-3-butyric acid (IBA). The concentration of BAP at 1.0 mg/l showed the highest survival rate (70%) followed by 0.5 mg/l then the control treatment. Similarly, concentration of BAP at 1.0 mg/l produced a higher number of responded explants (2.66) and shoot number (2.67) compared with the other treatments. In multiplication media BAP at 2.0 mg/l without NAA produced higher percent of responded shoots; the lower concentrations of BAP gave lower response. The highest shoot number was observed into multiplication medium supplemented with BAP at 2.0 mg/l then by BAP at 2.0 mg/l +0.2 mg/l NAA. Meanwhile, shoot length showed a different trend in this experiment, as the highest shoot length occurred at the control treatment (0.0 BAP +0.0 NAA) followed by all BAP treatments, while addition of NAA to BAP into multiplication medium gave lower shoots length. Juniper shoots are hardly to root as, most the treatments were inefficient. OM medium was responsible for rooting only when addition of IBA was implemented. The maximum percentage of rooted shoots was obtained with olive medium supplemented with IBA at 1.0 mg/l. According to the obtained results Juniper is a recalcitrant species to *in vitro* conditions; the multiplication rate highly depends on BAP concentration. Also, Juniper shoots have a low rooting potential, and most of the examined treatments were inefficient. Future studies are required to improve the current *in vitro* propagation potential.

Corresponding author
Alae Ahmed Jabbour,
alaeperesearch@hotmail.com

## INTRODUCTION

The second most common genus of conifers on Earth is *Juniperus*, which belongs to the Cupressaceae family. The *Juniperus* genus, comprising over 75 species, is one of the most widely distributed coniferous genera globally. Many species within this genus are found in the Northern Hemisphere, Africa, Central America, Europe, and Asia. Some species are capable of thriving in arid regions and adapting to harsh environmental conditions. One such species, *Juniperus procera*, known as 'Arar' in Arabic, grows in the Enemas region of southern Saudi Arabia. It is a coniferous, evergreen shrub or tree species Arabia (*Loureiro et al., 2007*; *Seca, Pinto & Silva, 2015*; *Hazubska-Przybył, 2019*). Coniferous evergreen shrub or tree is called *Juniperus procera*; *Juniperus* comprises more than 75 species (*Loureiro et al., 2007*). *Juniperus procera* leaves, fruits and seeds are rich with bioactive compounds, with high anticancer, antimicrobial, and antioxidant activities (*Nuñez et al., 2008*; *Tumen et al., 2013*; *Abdel Ghany & Hakamy, 2014*; *Bitew, 2015*). According to *Mujwah, Mohammed & Ahmed (2010)*, *J. procera* leaves are a source of novel flavonoids. In addition, fruits can be used medicinally to treat skin conditions and headaches. Its resin was used as a stimulant and medication to treat liver disorders and ulcers when combined with honey (*Jansen, 1981*; *Ahani et al., 2013*; *Tounekti, Mahdhi & Khemira, 2019*). Nevertheless, juniper forests are disappearing from many of today's woods because of both human activities and natural causes. This species has been steadily diminishing in many parts of the world; mostly due to drought, soil erosion, and increased runoff (*Ortiz, Arista & Talavera, 1998*; *El-Juhany, 2015*; *Aref et al., 2016*). Additionally, the distribution of populations of many juniper species may have been negatively impacted by climate change. Because of this, certain juniper species are currently considered rare or endangered and need to be protected right away (*Romme et al., 2009*; *Abrha et al., 2018*; *Khater & Benbouza, 2019*). The development of efficient *ex situ* conservation techniques, reproduction for upcoming reintroduction and restoration projects is therefore critically needed (*Mestanza-Ramón et al., 2020*). Traditional forestry reintroduction procedures as seed propagation, rooted cuttings, and grafting have been employed with conservation strategies (*Varshney & Anis, 2014*). With the ability to conserve and mass clonally propagate many coniferous tree species, *in vitro* culture technology is becoming more and more popular (*Lynch, 1999*). Certain species of *Juniperus* have either produced extremely few viable or anatomically underdeveloped seeds, moreover high percentage of empty seeds (*Ortiz, Arista & Talavera, 1998*; *Mohammadi Zade et al., 2018*), in addition to mechanical dormancy and presence of germination inhibitors (*Juan et al., 2006*). As a result, *in vitro* culture technique is gaining more and more attention as a potential substitute for mass clonal propagation and conservation of many coniferous tree species (*Hazubska-Przybył, 2019*). *In vitro* propagation techniques are useful for mass production, conservation and restoration of forestry trees and overcome the problem of deforestation (*Zaidi et al., 2012*). On the other hand, plant *in vitro* propagation can solve the issues with conventional conservation methods because it is affected by the environment conditions, ageing of the plants, diseases and pests (*Panis, Nagel & Van DenHouwe, 2020*). As a result, it has proven to be a trustworthy technique for plant multiplication, particularly when it comes to producing uncommon and endangered species (*Francis, Senapati & Rout, 2007*;

*Joshi & Dhawan, 2007*; *Offord & Tyler, 2009*; *Gonçalves, Fernandes & Romano, 2010*). The first research on the *in vitro* propagation of juniper associates carried out by *Javeed, Perveen & Ilahi (1980)*. On the topic of juniper *in vitro* propagation, a few investigations have been conducted and published (*Zaidi et al., 2012*; *Gomez & Segura, 1994*; *Gomez & Segura, 1995*; *Negussie, 1997*; *Khater & Benbouza, 2019*). Because micropropagation may be the sole other means of plant reproduction for this group, *in vitro* propagation of juniper species should be overstated (*Hazubska-Przybył, 2019*). Prioritizing it would be beneficial for both mass propagation for pharmaceutical applications and the possibility of conservation (*Harry, Pulido & Thorpe, 1995*). Plant tissue culture is an effective way to increase the production of secondary metabolites (*Hussain et al., 2012*). Because *J. procera* regeneration through seeds is difficult, bulk propagation material for pharmaceutical purposes can be produced using the micropropagation technology, which will preserve natural regeneration through seeds in the wild. Consequently, the main goal of the current experiment was to optimize the medium and plant growth regulator (PGRs) concentrations for *in vitro* proliferation of *J. procera* shoots and roots to overcome the issues with vegetative reproduction and seed regeneration.

## MATERIALS AND METHODS

### Plant material and explant sterilization

Shoots of selected Juniper trees (*Juniperus procera*), were collected from Baljurashi, Al-Baha region (41°35′4″E, 19°50′53.7″N) on the western south of Kingdom of Saudi Arabia and immediately transferred to the tissue culture laboratory, shoots were divided into segments each included 2–3 nodes.

Shoot segments were washed under running water for half an hour then rinsed with alcohol at 70% for 10 s. Four sterilization treatments were processed using Clorox (commercial bleach containing 5.5% of NaOCl) and mercuric chlorate (MC) as the following:

T1 - Clorox at 50% for 15 min followed by MC at 0.2 g/l.
T2 - Clorox at 50% for 30 min followed by MC at 0.1 g/l.
T3 - Clorox at 50% for 30 min followed by Clorox at 30% for 15 min.
T4 - Clorox at 50% for 30 min followed by Clorox at 30% for 30 min.

Explants were rinsed three times with distilled sterilized water and cultured on autoclaved Murashige and Skoog (MS) media (*Murashige & Skoog, 1962*), supplemented with benzyl amino purine (BAP) at 0.5 mg/l, sucrose at 30 g/l and agar at 8.0 g/l, cultured explants were incubated at growth chamber ($25 \pm 2$ °C and 16 h photoperiod) for three weeks. Survival rate and contamination percentage were recorded.

### Initiation stage

Juniper sterilized explants were cultured in MS media, supplemented with sucrose at 30 g/l and agar at 8.0 g/l and BAP at 0.0, 0.5 and 1.0 mg/l for shoot initiation. Cultured explants were incubated under growth chamber conditions (16/8 h (day/night) photoperiod at intensity of 1,500 lux and $23 \pm 2$ °C). Nine jars each contained one explant were used for

Table 1 BAP and NAA concentration during multiplication stage of Juniper shoots.

| Treatments | BAP | NAA |
|---|---|---|
| T1 | 0.0 | 0.0 |
| T2 | | 0.0 |
| T3 | 0.5 | 0.2 |
| T4 | | 0.4 |
| T5 | | 0.0 |
| T6 | 1.0 | 0.2 |
| T7 | | 0.4 |
| T8 | | 0.0 |
| T9 | 2.0 | 0.2 |
| T10 | | 0.4 |

each treatment. Four weeks later, survival rate, number of sprouted buds and number of shoots per explant were recorded.

## Multiplication stage

The sprouted Juniper explants from initiation media were sub-cultured into multiplication medium. MS medium supplemented with BAP at 0.5, 1.0 and 2.0 mg/l with naphthaleneacetic acid (NAA) at 0.0, 0.2 and 0.4 mg/l in addition to the control treatment (MS free growth regulators) were used for multiplication rate assessment (Table 1). Nine jars each contained two shoots were used for this treatment. Sprouted shoots percentage, average shoot number per explant, average leaf number and shoot length were estimated.

## Elongation stage

Shoot from the above mentioned treatments was transferred to $\frac{3}{4}$ MS + 0.5 mg/l kinetin + 1.0 mg/l Indole-3-acetic acid (IAA) as described by *Taha et al. (2021)*. Two subcultures exploited this medium (4-weeks interval). Cultures were incubated at 23 ± 2 °C and 2,000 lux of light intensity before transfer to rooting media.

## Rooting stage

*In vitro* growing Juniper shoots with length of 2–3 cm taken from elongation medium were cultured in half strength MS or full strength olive medium (OM) (*Rugini, 1984*), supplemented with indole-3-butyric acid (IBA) at 0.0, 0.5, 1.0 and 1.5 mg/l. Ten shoots were cultured for each treatment and incubated at growth chamber at 16/8 h (day/night) photoperiod and intensity of 3,000 flux. Plantlet length, rooting percentage, root number and root length were determined.

## Statistical analysis

The treatments of the current research were arranged in complete randomized design, with three replicate (three jars with three explants in each jar) for each treatment. Data normality was tested using Shapiro–Wilk's test, percentage data were subjected to square root transformation to ensure that the data follow normal distribution pattern. Statistical analysis was performed with normalized data, but all results are shown as original data.

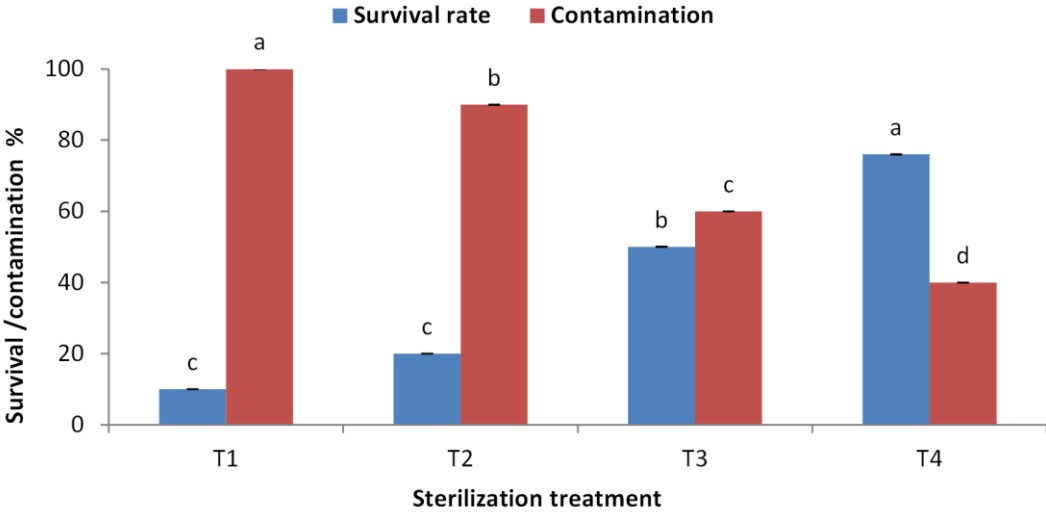

**Figure 1** **Effect of sterilization agent, concentration and duration on *Juniperus procera* explants; different letters indicate statistical differences between treatments according LSD test ($p \leq 0.01$).** Error bars represent the standard deviation. T1; Clorox at 50% for 15 min. + by MC at 0.2 g/l, T2; Clorox at 50% for 30 min. + by MC at 0.1 g/l, T3; Clorox at 50% for 30 min + by Clorox at 30% for 15 min, T4; Clorox at 50% for 30 min + by Clorox at 30% for 30 min.

Data were subjected to one way analysis of variance (ANOVA) to investigate the effect of nutrient media or hormonal concentration on the recorded parameters. Analysis of variance was performed using MSTAT-C statistical package software (*Freed et al., 1990*). Significant differences ($p \leq 0.01$) were determined using least significant distance (LSD) test (*Snedecor & Cochran, 1967*). The mean and standard deviation ($\pm$ SD) were calculated from three replicates per treatment.

## RESULTS

### Effect of type of sterilization agent, concentration and duration on explants

The highest survival rate (80%) for juniper explants was recorded when Clorox was used at 50% then 30% each for 30 min. followed by the same concentrations but for 30 min and 15 min, respectively (Fig. 1). The lowest survival rate was recorded when Clorox was used at 50% for 15 min then explants were immersed in MC at 0.2 g/l. This later treatment also showed a dramatically high contamination level. MC showed a toxic effect on juniper explants and did not decrease the contamination levels to a satisfied level. Fortunately, treating explants with Clorox at 50% then 30%, each for 30 min, lowered the contamination to the lowest level (40%).

### Effect of BAP concentration on initiation stage of explants

The obtained results indicated that BAP is crucial for sprouting of juniper explants. The concentration of 1.0 mg/l showed the highest sprouting rate (70%) followed by 0.5 mg/l compared with the control treatment (Fig. 2). Similarly, the concentration of 1.0 mg/l

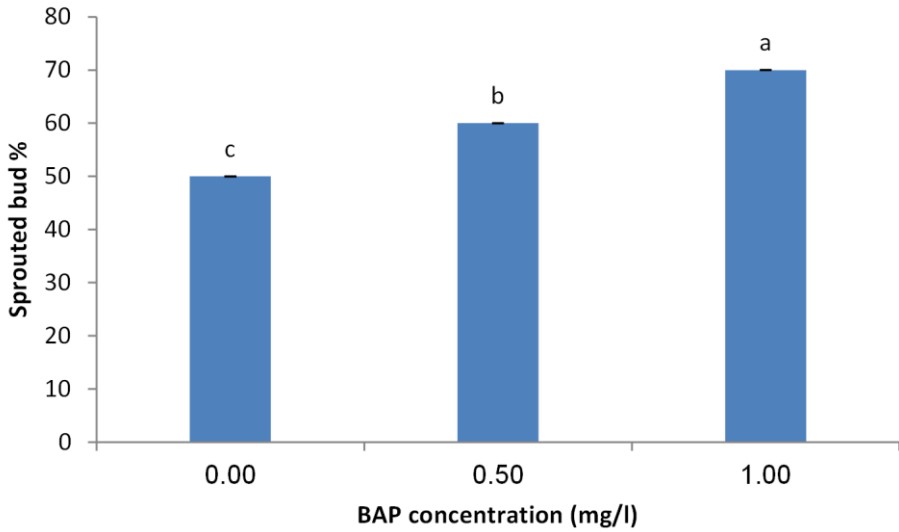

**Figure 2** **Effect of BAP concentration on sprouting percentage of *Juniperus procera* explants; different letters indicate statistical differences between treatments according LSD test ($p \leq 0.01$).** Error bars represent the standard deviation.

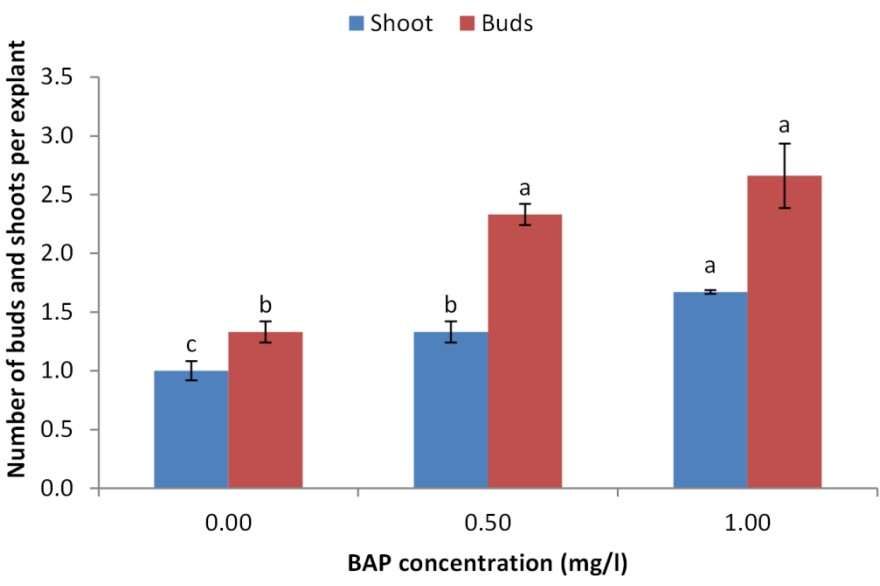

**Figure 3** **Effect of BAP concentration on number of sprouted buds and shoots per explant of *Juniperus procera*; different letters indicate statistical differences between treatments according LSD test ($p \leq 0.01$).** Error bars represent the standard deviation.

obtained the highest number of sprouted buds per explants. Average number of shoots per explants also took the same trend as the concentration of 1.0 mg/l showed the highest shoot number (Fig. 3).
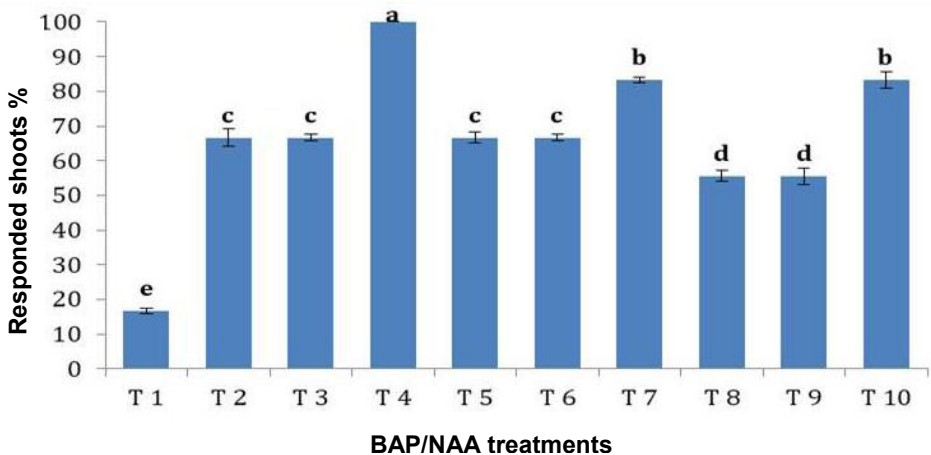

**Figure 4** **Effect of BAP and NAA combination on responded shoots percentage of *Juniperus procera*; different letters indicate statistical differences between treatments according LSD test ($p \leq 0.01$).** Error bars represent the standard deviation; T.

## Effect of BAP and NAA concentrations on multiplication stage of shoots

BAP had a significant effect on multiplication rate of juniper shoot. The highest percentage of responded shoots was occurred with 2.0 mg/l BAP without NAA while the lower concentrations of BAP gave lower response (Fig. 4). NAA combination with BAP, lower the percentage of responded shoots. The best results with addition of NAA were noticed when combined with BAP at 2.0 mg/l. With respect to shoot number parameter, data in Fig. 5 showed that the highest shoot number was observed into medium supplemented with BAP at 2.0 mg/l followed with BAP at 1.0 mg/l and 2.0 mg/l BAP+0.2 mg/l NAA (Figs. 6A–6C). The higher concentration of NAA negatively affected shoot number. Meanwhile, shoot length showed a different trend as the highest shoot length was occurred at the control treatment (0.0 mg/l BAP +0.0 mg/l NAA) followed by all BAP treatments at 0.5 mg/l while, the higher BAP concentrations (1 and 2 mg/l) recorded lower shoot length (Fig. 7), the effect of NAA was inconstant. Leaf number had the same trend with shoot length as the control treatment gave the highest leaf number of juniper shoots. BAP at 1 or 2 mg/l recorded lower leaf number; NAA combined with higher BAP concentration recorded higher leaves number compared with BAP alone (Fig. 8).

## Effect of nutrient media and IBA concentrations on rooting stage of shoots

Shoots from different multiplication media were transferred to $\frac{3}{4}$ MS supplemented with 0.5 mg/l kinetin and 1.0 mg/l IAA. After two subcultures the cultured shoots showed enhanced shoot length and become suitable to sub-culture on the rooting media (Figs. 6D and 6E). Shoots (2–3 cm in length) produced from elongation medium were selected to use in rooting experiment. Data assured that juniper shoots are hard to root; most of the examined media and IBA concentration were inefficient to induce root formation

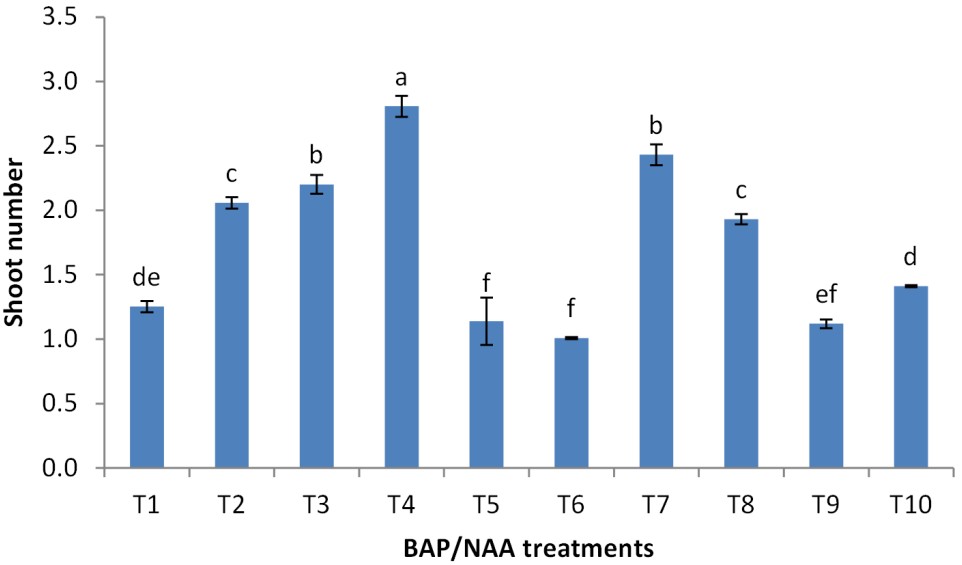

**Figure 5** **Effect of BAP and NAA combination on shoot number of *Juniperus procera*; different letters indicate statistical differences between treatments according LSD test ($p \leq 0.01$).** Error bars represent the standard deviation; T1 to T10 refer to BAP.

on *Juniperus* micro-shoots. All juniper shoots cultured on $\frac{1}{2}$MS media with all IBA concentration failed to root, OM medium was responsible for rooting only when addition of IBA was implemented; Olive medium supplemented with IBA at 1.0 mg/l gave the highest percentage of rooted shoots (Figs. 9 and 10). With respect to shoot length, the highest value was occurred at $\frac{1}{2}$ MS+1.0 IBA and OM+1.0 IBA followed by $\frac{1}{2}$ MS+0.5 IBA and OM+0.5 IBA or 1.5 IBA (Fig. 11). With respect to leaves number, the highest value was occurred at $\frac{1}{2}$ MS+1.0 IBA and OM+1.0 IBA followed by $\frac{1}{2}$ MS+0.5 IBA then OM+0.5 IBA (Fig. 12).

## DISCUSSION

*In vitro* culture technique has potential substitute for mass clonal propagation of many tree species (*Hazubska-Przybyl, 2019*). Contamination with different microorganism is a serious problem, of micro-propagation; eliminate microbial contamination is a basic requirements for establishment of cultured plant tissues. Our results indicated that Clorox (5% sodium hypochlorite) is good sterilization agent for juniper explant and the treatment for 30 min with Clorox at 50% followed by 30 min with Clorox at 30% gave the highest survival rate and the lowest contamination percentage. The sterilization efficiency of sodium hypochlorite was reported previously, for example, *Juniprus navicularis* micro-cuttings were sterilized using 70% ethanol, 3% commercial bleach, and Benlate solution 1% (*Castro et al., 2011*). Also, *J. excelsa* shoot tip explants showed higher sterilizing when treated with 2.5% sodium hypochlorite (*Kashani et al., 2018*). A high sterilized degree was obtained by using commercial bleach (1–5%), and HgCl$_2$ (0.1–1%) according to explant age, size, and source of the explants (*Darwesh, Zaied & Hassan, 2024*). Immersing of explant

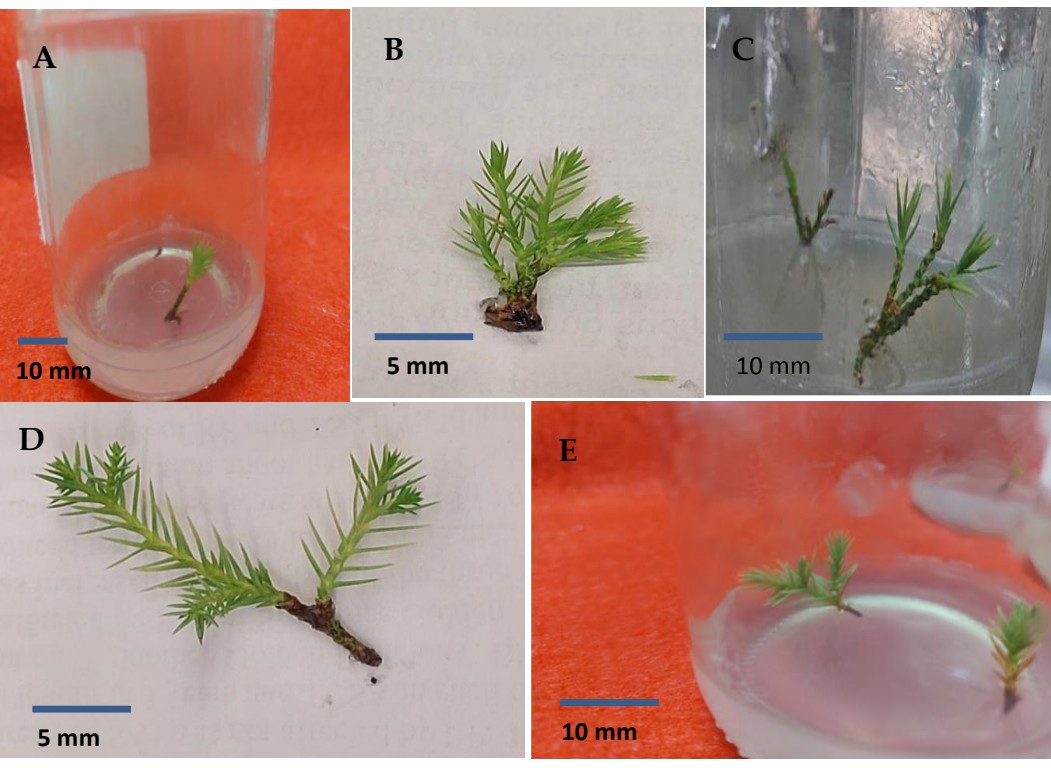

**Figure 6** Sprouted juniper shoots (A); multiple shoots produced on MS medium supplemented with 2.0 mg/l BAP (B and C); elongated shoots cultured on 3/4 MS medium with 0.5 mg/l kinetin + 1.0 mg/l IAA (D and E).

in a fungicide solution, followed by immersion in NaOCl solution, a very high level of sterilization was obtained (*Khater & Benbouza, 2019*). Sodium hypochlorite treatment resulted in higher survival percentage for pomegranate explants (*Singh et al., 2010*). Using a combination of NaOCl successfully sterilizes axially bud and stem segments in pomegranate and Jack fruit with good survival rate were obtained (*Damisno, Padro & Frattarelli, 2008*; *Faisal et al., 2010*). Shoot multiplication is highly affected by plant genotype, growth medium, and cytokinin; our results indicated that BAP was essential for proliferation and multiplication of juniper explant. Similarly, *Qarachoboogh et al. (2022)* found that the optimal culture medium for shoot growth of *J. foetidissima* was MS supplemented with BAP at 1.0 mg/l and IAA at 0.1 mg/l, while *Salih et al. (2021)* assured that the highest shoot multiplication was obtained with 0.5 µM BAP combined with 0.5 µM IAA or 0.5 µM IBA. Higher shoots number was produced in medium containing 0.5 mg/l BAP as *J. excelsa* produced six shoots, *J. horizontalis* produced eight shoots and *J. chinensis* produced nine shoots per explant. However, presence of auxin maybe inefficient or retard shoots growth as we noticed in the current experiment; NAA had a negative effect on shoot number and shoot length. *Kuritskaya, Vrzhosek & Boltenkov (2016)* assured that when the IBA concentration was raised in the medium of *J. chinensis* var. sargentii the number of buds and the length of shoots reduced. In addition, 0.5 mg/L of BAP resulted in a greater shoot

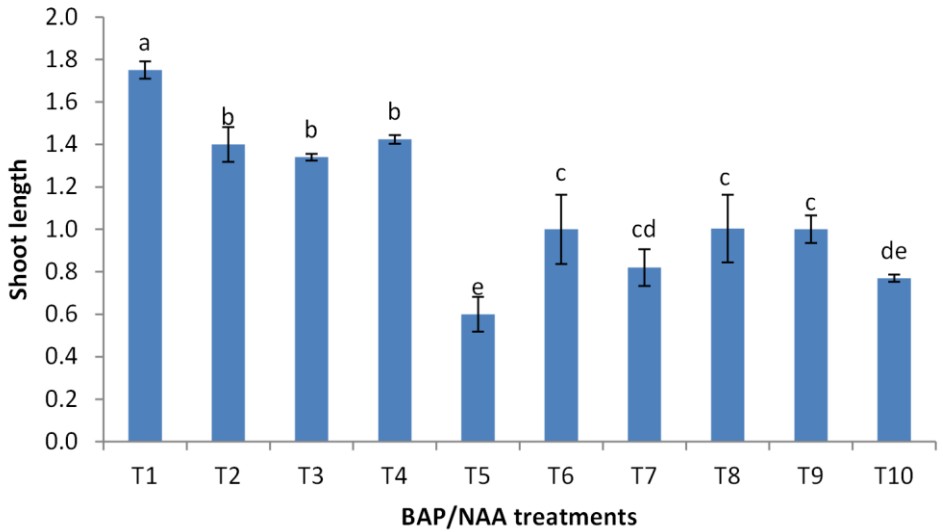

**Figure 7** Effect of BAP and NAA combination on shoot length of *Juniperus procera*; different letters indicate statistical differences between treatments according LSD test ($p \leq 0.01$). Error bars represent the standard deviation; T1 to T10 refer to BAP.

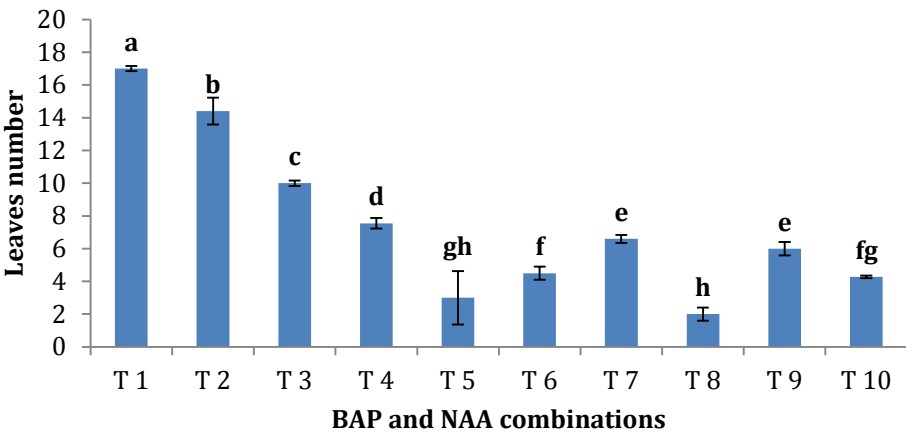

**Figure 8** Effect of BAP and NAA combination on leaves number of *Juniperus procera*; different letters indicate statistical differences between treatments according LSD test ($p \leq 0.01$). Error bars represent the standard deviation; T1 to T10 refer to BA.

proliferation rate (5.37 shoots per explant) of a dwarfing cherry rootstock (*Mahdavian, Bouzari & Abdollah, 2011*). In addition, sweet cherry cultivar "Lapins" demonstrated high frequency shoot proliferation when grown on the basic MS medium with reduced BAP concentrations. Conversely, a greater concentration of BAP also produced good shoot elongation (*Ruzic & Vujovic, 2008*). The highest shoots number of Jack fruit explant was obtained on MS medium with BAP (*Damisno, Padro & Frattarelli, 2008*). Many researchers tried to produce juniper rooted plantlets but it seems that juniper is hard to root. *Ioannidis*

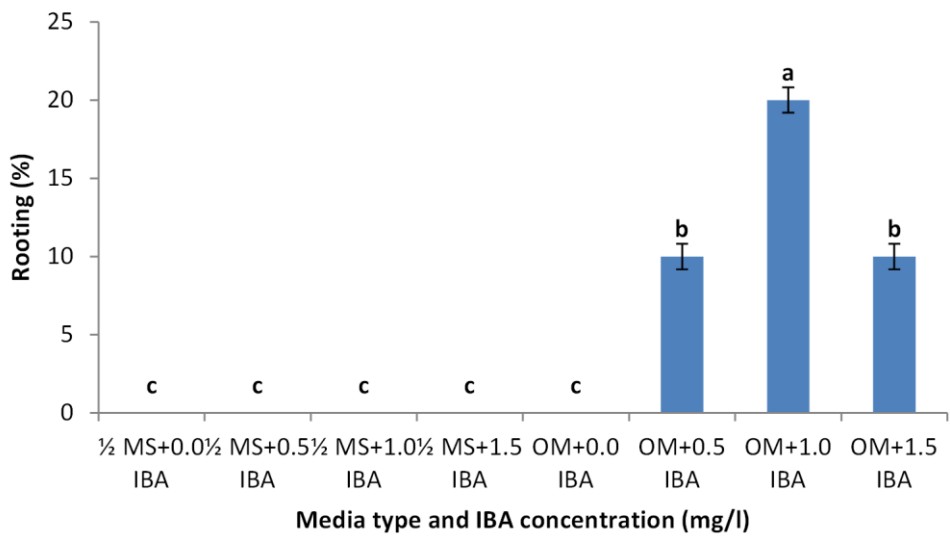

**Figure 9** Effect of media type and IBA concentrations on rooting percentage of *Juniperus procera* shoots; different letters indicate statistical differences between treatments according LSD test ($p \leq 0.01$). Error bars represent the standard deviation.

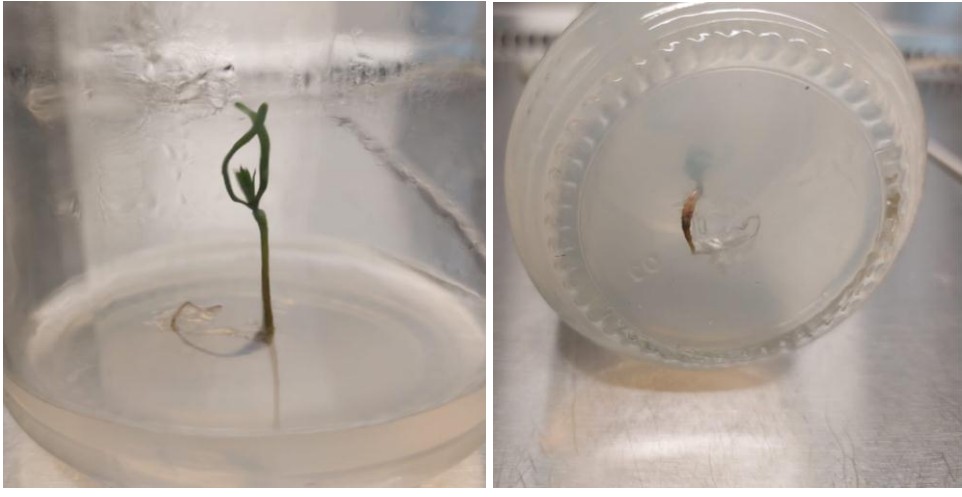

**Figure 10** Rooted *Juniperus procera* shoots growing on OM supplemented with IBA at 1 mg/l.

*et al. (2023)* studied *Juniperus drupacea*; they found that using IBA, NAA, or IAA in various concentrations was proven to be ineffective for its rooting. According to *Qarachoboogh et al. (2022)* and *Güney et al. (2021)* juniper is rarely propagation by cuttings due to the poor rooting of stem cuttings; which may be overcome by *in vitro* rooting. Root induction depends on the composition of culture media and phytohormones; moreover, auxin concentration and method of treatment are important factors affecting root induction (*Amiri et al., 2019*). IBA is the commonly applied at low concentrations for root induction

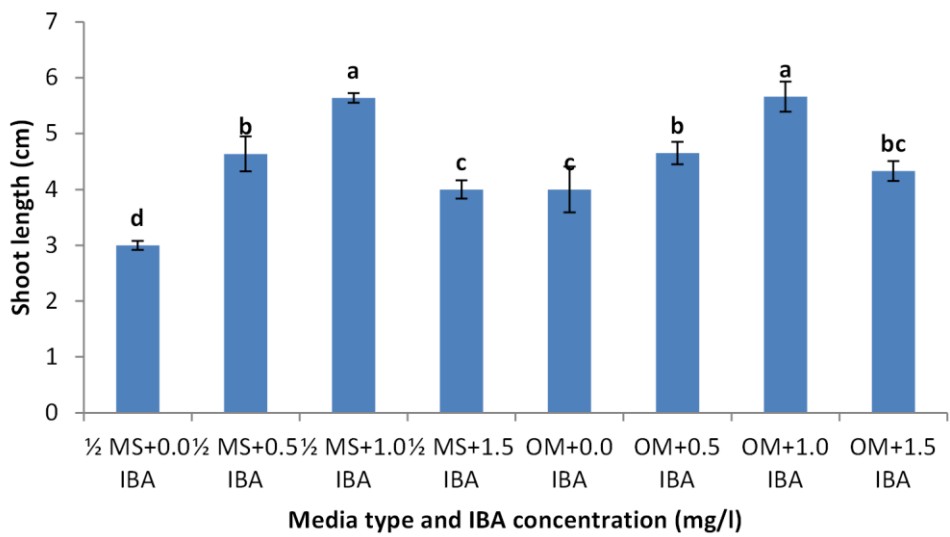

**Figure 11** Effect of media type and IBA concentrations on shoot length of *Juniperusprocera* shoots; different letters indicate statistical differences between treatments according LSD test ($p \leq 0.01$). Error bars represent the standard deviation.

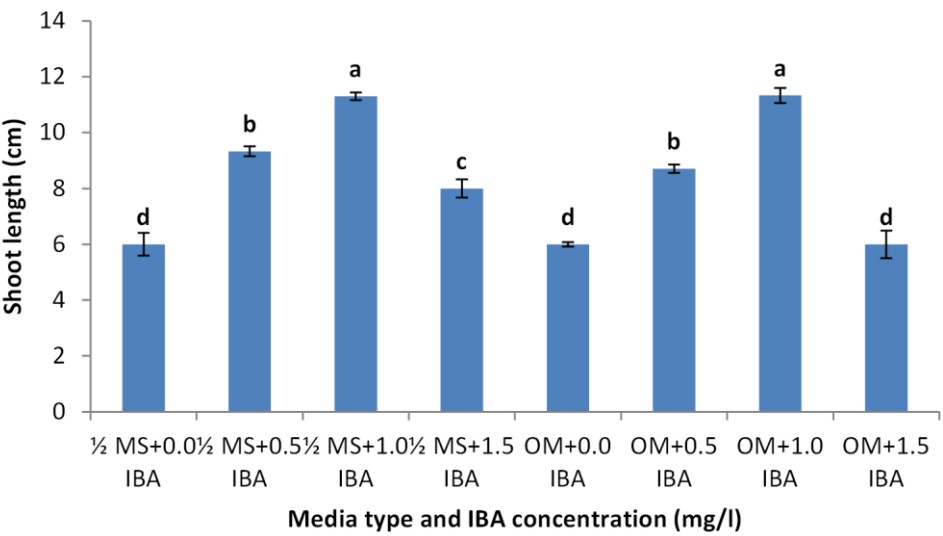

**Figure 12** Effect of media type and IBA concentrations on leaves number of *Juniperusprocera* shoots; different letters indicate statistical differences between treatments according LSD test ($p \leq 0.01$). Error bars represent the standard deviation.

(*Beyramizadeh, Arminian & Fazeli, 2020*), further increase in auxin concentration inhibited rooting growth (*Negussie, 1997*). *Farzan, Rezanejad & Zamanid (2023)* reported that root formation of juniper shoots was not observed until six months, but about 10% of these regenerated shoots produced roots eight months after shoot proliferation. They claimed that various factors such as genotype, polyploidy or hybrid formation, slow plant growth

(which is common in conifers), or secondary compounds are the reasons for the low rate of regeneration, especially rooting. Nevertheless, rooting occurred in three species of *Juniperus* using shoots with length of 4–7 cm. The best roots produced in woody plant medium (WPM) with IBA at 1.0 mg/l but these roots are not enough to support plantlet (*Zaidi et al., 2012*). Similarly, *Castro et al. (2011)* reported that more rooting was obtained in juniper micro-shoots cultured on OM supplemented with IBA at 12.3 µM. However, *Negussie (1997)* indicated that spontaneous rooting at a low percentage (10.0%) could be observed on WPM media with 0.1 mg/l IBA after a long period of cultures, further increase in IBA concentration inhibited root growth. In addition, *Kuritskaya, Vrzhosek & Boltenkov (2016)* assured that adventitious roots developed in *Microbiota decussate* after three months of cultivation on MS medium with 0.1 mg/l IBA. Nevertheless, *Kocer et al. (2011)* said that adventitious root-like structures were formed in multiple experimental trials using 0.005, 0.03, and 0.05 mg/l of IBA; however, none of these structures were differentiated into a real root system. Rugini medium has enriched composition compared to MS and contains folic acid which was found to be useful in root induction (*Mustafa et al., 2018*). OM supplemented with three mg/l IBA achieved rooting percentage of 85% (*Peixe et al., 2007*). Also, MS medium containing 2.0 mg/l IBA has been found essential for obtaining good rooting during *in vitro* rooting of pomegranate (*Singh et al., 2010*).

## CONCLUSIONS

Generally the obtained results indicated that Juniper is a recalcitrant species to *in vitro* conditions; the micropropagation of Juniper is highly dependent on the nutrient media and growth regulators. The highest proliferation rate and shoot number was recorded for BAP at 2.0 mg/l L-1; addition of NAA negatively affected multiplication rate and shoots growth. Juniper shoots demonstrated a low rooting potential, as most of the examined treatments were inefficient; addition of 1.0 mg/l IBA to olive media exhibited better results than the other treatments. Future studies are required to improve the current micropropagation protocol.

## ACKNOWLEDGEMENTS

The authors would like to express special gratitude to the Faculty of Applied Science, Umm Al-Qura University, and Faculty of Science, Al-Baha University, for the use of its facilities during this research.

### Funding
The authors received no funding for this work.

### Competing Interests
The authors declare there are no competing interests.

## Author Contributions

- Alae Ahmed Jabbour conceived and designed the experiments, performed the experiments, analyzed the data, prepared figures and/or tables, authored or reviewed drafts of the article, and approved the final draft.
- Abdulaziz Alzahrani performed the experiments, analyzed the data, authored or reviewed drafts of the article, and approved the final draft.

## Data Availability

The raw data is available in the Supplementary File.

## Supplemental Information

Supplemental information for this article can be found online at http://dx.doi.org/10.7717/peerj.19255#supplemental-information.

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
