# Peer review of "Efficiency improvement of Juniper trees mass propagation through in vitro shoot multiplication"

_PeerJ, doi:10.7717/peerj.19255_

## Round 0.1 · original submission · Major Revisions

Authors should take into consideration the comments of the reviewers who specifically suggest improvements in the clarity of the manuscript and its scientific rigor.

Reviewer 1 ·

Basic reporting

The writing is clear. But, the consistency of the writing style is poor. Sometimes the sentences are in passive forms, but sometimes in active forms.
Line 77 - species name should be in italic font

Experimental design

The treatments were not designed in proper manners. There is no control and poor design. Example: Line 92 and Line 93 - why the duration is not fix at 30 minutes for each concentration? so the author will not know whether the duration or the concentration that influence? as both were not systematically designed.

Validity of the findings

The results are in doubt. Example: Survival / contamination percentage - How actually the authors did the statistical analysis? bar graph that is higher than the other bar is written as 'c' and the lower bar is 'b'?
Data on shoot length, the bar for bc, c, cd, are not convincing. I believe the error in the analysis is high - maybe due to the number of replications is too less / the number of units per replicate is too low.

Basically, all the data is not convincing, based on the graphs. The authors should also mention the coefficient variation in each analysis.

Additional comments

The experiment was done in not systematic method and the statistical analysis should be done properly (if the data is correct.

Reviewer 2 ·

Basic reporting

Description of results needs rewriting

Experimental design

Methods require minor additions

Validity of the findings

no comment

Additional comments

The manuscript titled “Effect of Media Composition and Growth Regulators on Mass propagation of Juniper (Juniperus Procera)” by Jabbour and Alzahrani presents the results of optimizing the micropropagation procedure of a medicinal, rare plant prone to dieback and difficult to propagate by seed. The effects of MS and OM (olive medium) and the plant growth regulators BAP, NAA and IBA on the performance of the various stages of microshoots growth in in vitro culture were tested. The highest intensity of microshoot multiplication was obtained in the presence of BAP at a concentration of 1.0 mg/L added alone to MS medium. Shoot length was highest on media without plant growth regulators, and the frequency of shoot rooting in the presence of IBA added to OM, at a concentration of 1.0 mg/l. The results presented in the paper indicate that the micropropagation of J. procera is promising alternative method, although still the rooting stage, which is a very difficult step in juniper, requires further studies.
I believe that the presented results are worthy of publication, especially since research conducted on the use of micropropagation techniques in the propagation of species of the genus Juniperus is still not widespread. Nevertheless, the reviewed manuscript still requires numerous corrections and additions to qualify for publication. Below are a number of comments and suggestions to the authors in order to improve the quality of the text and the readability of the analyses presented.

Title


In my opinion, the title is stiff, not very informative. Therefore, I suggest to change it in this way, for example:

„Efficiency of Juniperus procera micropropagation through shoot multiplication”

Abstract

The abstract should be rewritten. The abstract should be rewritten, according to the scheme:
1. The purpose of the research, questions, thesis or hypotheses;
2. The research methods
3. The main findings/results
4. Conclusions, which can be drawn from the study

Each part should consist from 1-2 sentences.

For example, start like this:
“Juniperus procera is an endangered medicinal tree with limited reproductive capabilities, so reaching for alternative propagation methods based on in vitro techniques could help save this species of valuable plant......”

Note: the Methods section should list the two media tested in order to maintain continuity in the presentation of results later in the abstract. The abstract is missing a conclusion.
I also suggest rewording the keywords so that the words from the title are not repeated here

Key words: Tissue culture; African juniper; BAP, Shoot rooting, Medicinal Endangered plant

Note for the future regarding the manuscript: Please number the pages, which makes it easier for the reviewer to navigate through the text.

1. Introduction

Page 1, lines 35-36
Juniperus procera; use italics throughout the manuscript for names of species, genera et..,

Page 2, lines 37 and 39
Standardize the notation of the phrase “et al.” here and throughout the manuscript.

Page 2, line 42
According to References the work Mujwah 2010 has much more co-authors

Page 3, line 80
promethea? Is it the another name of P. procera?

Page 3, line 83
Add to the sentence the word „plant” before growth regulators and abbrevation (PGRs)
Use abbreviation in the whole manuscript text, consequently.


2. Materials and methods
2.1 Plant material and explant sterilization

Page 3, lines 89-90
Standardize the notation of the phrase “Figure, 1” (with or without comma in parentheses) here and throughout the manuscript. Change „lab.” into „laboratory”.

Page 3, line 93
Add information about the sterilization time of the shoot segments in alcohol.

2.2 Initiation stage
Page 4, line 106
(…) for initiation. (…) change to (…) for shoots initiation. (…)

Page 4, line 107
Describe the growth chamber conditions in this way:
a 16/8 h (day/night) photoperiod at (…) and the illumination/light (? add the correct one) at an intensity of (…).

flux or lux?

2.3 Multiplication stage

Page 4, line 111
Remove „and” from the sentence.

Page 4, lines 112-114
It would be useful to add an additional table or list the PGR combinations in the text to facilitate understanding of the experiment model.

2.4 Elongation stage

Page 4, line 118
The phrase „(…) therapies was moved to (…)” is clumsy, instead, it would be better to write „treatments have been transfered to”

Page 4, line 120
Instead of „(…) is required for (…)” write „(…) before (…)” to improve the sentence.

2.5 Rooting stage

Page 4, lines 123-124
Add the missing word „taken” before „(…) from elongation medium (…)” and abbreviation „OM” in the parentheses before the name of the author of the medium).

Page 4, line 126
Give the length of the photoperiod as previously suggested by the reviewer.

2.6 Statistical design (maybe better would be analysis?)

Page 4, line 131

Correct the sentence: „(…) Data were data was subjected (…)”

3. Results

General comments on all subsections of this chapter:

The discussion of the results should begin with a description of what was observed in vitro culture (how the microshoots looked and developed), and then we refer to the results of the measurements and statistical analyses performed. In contrast, the authors begin discussing the results in each chapter by referring to the tables, which should be corrected. After presenting the results, a reference to a specific table should be included in the first sentence so that readers can immediately follow the results as they read the text.

3.1 Effect of type …

Pages 4/5, lines 137-139

I suggest to begin this chapter in this manner:

„The highest survival rate (80%) for juniper explants (…) (Figure 2).The lowest (…)”
By the way, fix „(…) Figure (2) (…)” for „Figure 2” here and use the same manner throughout the manuscript text. Moreover, description of the results, should follow the order of presentation in the figure, so the variant of treatment for 15 min with 30% Clorox should be listed first.

Page 5, line 140
Instead of „mercuric chloride” use abbreviation „MC”. The same applies to other names and abbreviations used. Attention should be paid to.

Page 5, lines 144-145
This sentence is unclear.

3.2 Effect of BAP concentration …

Page 5, lines 147-149
I suggest to begin this chapter in this way:

This experiment revealed that (…) (Figure 3). The highest sprouting rate (70%) was obtained after (…)”

Page 5, line 150
Explain whether the number or average number of sprouted buds/shoots per explant was taken into account in the experiment.

3.3 Effect of BAP concentration …
Improve the subsection according to rewiever's earlier suggestions.
Page 5, lines 166-169
The results presented are shown in Table 8 and not Table 6, as the authors state in the last sentence. And the finding that the higher the concentration of BAP the lower the number of leaves is not entirely consistent with the analytical results presented.

Page 5, line 172
What does it mean „the profiled shoots”?

Add the name of subchapter 3.4 concerning the rooting of the shoots. It was missed. Correct also this subchapter according to the rewiever’s previous suggestions.

Page 6, lines 182-186
Figure 13 should be renumbered to 11, as it is mentioned earlier in the text. Automatically Figure 11 will be changed to 12 , and 12 to 13. These changes should also be taken into account when referring to the results in the discussion.

Page 6, lines 183 and 185
½ MS +1.5 IBA, there is a mistake. According to cited figures it should be 1.0 IBA.

4. Discussion

Page 6, lines 188-196
This part of the text is redundant here and should be found in the Introduction. The chapter could begin with a statement of the purpose for which the research was undertaken, or it could begin immediately with an interpretation of the key findings.

Page 6, line 197
Complete the sterilization time in Clorox solutions in parentheses.

Page 6, line 197
Part of the sentence: „(…) was reported previously Juniperus navicularis microcuttings was (…)” is unclear. I suggest changing it this way: „(…) was reported previously, for example for Juniperus navicularis, which microcuttings were (…)”

Page 6, lines 200 and 204
Give the year of publication of the works give in parentheses.

Page 6, line 206
Change „Sodium hypochloride recorded (…)” to „Sodium hypochloride treatment resulted in (…)”

Page 7, lines 208 and 226
Add „were obtained” after „(…) survival rate (…)” and remove „and” from parentheses.
Correct the cited literature „Damisno and Padro, 2008” to „Damisno et al. 2008”

Page 7, line 239
conferrals?

Page 8, line 256
For which plant species were these results obtained?
English needs language correction throughout the text. Numerous editorial shortcomings also need to be corrected, removing or adding dots, spaces, bolding, etc.

Figures

All figures require detailed descriptions. There are not included by the authors.
Figure 1
This figure is not of good quality and should be replaced if possible.

Figure 2
Provide the number of sterilization methods in the chart, according to the methods, and their descriptions under the figure to clarify. Also provide information on the statistical test, p-values, etc. On the Y-axis, replace the word “percent” with “%” and at the same time remove this symbol next to the individual values on the axis and from the legend description. Do the same for those graphs affected.

Figure 3
The reviewer suggests this description of the y-axis: Level of sprouted buds (%); on the x-axis, units should be in parentheses. Note this in the following graphs.

Figure 4
Shorten the legend description to buds and shoots, as this information is already included on the y-axis

Figures 5-7
The x-axis description should be corrected to read: BAP/NAA treatments and complete the descriptions of the options under the graphs

Figure 9
The letter „B” has disappeared from the figure.

Figures 11-13
Change the numbering of the figures, as previously suggested by the reviewer.

References

Page 9, line 230

The name of the species should be in italics, there are no spaces in the name and elsewhere in the bibliography. Comments refer to the entire list of literature.

Page 9, line 300
Bold the name of "Bitew" and the year of publication.

Page 10, lines 317-318 and 319-321
Swipping the two literature items.

Page 10, line 327
Incorrect publication year.

Page 11, line 378
The publication Murashige and Skoog 1964 is missing.

Page 13, line 431
Bold the year of publication.

·

Basic reporting

General Overview

The manuscript addresses the in vitro propagation of Juniperus procera, a significant species due to its medicinal properties and the challenges associated with its conservation. The study is relevant, and the methods and findings have the potential to contribute to the existing body of knowledge in plant tissue culture and conservation. However, several issues need to be addressed to improve the clarity, scientific rigor, and overall presentation of the manuscript.

Language and Clarity

The manuscript is generally written in clear and understandable English. However, there are grammatical errors and sentence structures that affect the flow. It would be beneficial for the authors to revise the language to enhance clarity.

Introduction and Literature Review

The introduction effectively highlights the significance of Juniperus procera and justifies the study. The literature review is sufficient.
The introduction provides a general overview of Juniperus procera and its conservation challenges. However, some information in the discussion section is redundant, as it has already been detailed in the introduction. The discussion should focus on interpreting the results, not repeating introductory details. Start the discussion with a concise summary of the study's scope and move directly into the interpretation of findings.

Figures and Tables

The figures are well-prepared overall, but some require clearer explanations. In addition, the resolution of some figures could be improved for better clarity.

Data and Visuals

The raw data are appropriately presented. Error bars are included in some result figures, but further details on ANOVA or the LSD test used should be provided alongside the results. Indeed, the manuscript states that ANOVA was performed, but the results are not clearly presented. To enhance clarity and reproducibility, the authors should include detailed ANOVA results, such as F values with their interpretations, p values with their significance levels, and the factors compared alongside an explanation of how variance was analyzed. Additionally, the LSD test results comparing group means should be presented either in a table or explicitly described in the results section. Providing these details will ensure better transparency and a more comprehensive understanding of the findings.

Experimental design

Research Question and Context

The research question is well-defined, aiming to optimize the micropropagation of J. procera. This objective addresses a clear gap in the existing literature and is relevant for both conservation and pharmaceutical applications.

Methods and Reproducibility

The methods are generally well-described, but some aspects lack sufficient detail. For instance:
• The rationale for selecting specific hormones and sterilization methods should be clearly explained. This would strengthen the study by demonstrating why these treatments were chosen over alternatives.
• The treatments (T1-T10) should be clearly outlined in the appropriate section to help readers understand the experimental design.
• Details about the ANOVA and LSD test methodologies and their interpretation should be clarified.

Ethical Compliance

There is no situation that poses an ethical concern.

Validity of the findings

Reliability of the Results

The findings are robust and generally support the study's hypothesis. The results of the study provide clear findings on the in vitro production of the species.

Statistical Validity

While the statistical analyses appear appropriate, the meaningfulness of the LSD test results is somewhat questionable. Additional statistical methods could be employed to strengthen the findings.

Linkage of Results to Conclusions

The results are well-linked to the research question and hypothesis. However, the section on future studies could be expanded to include recommendations for testing the effects of different combinations of growth regulators.

Discussion

The discussion section has been well-explained, supported by recent references. However, as noted in the text, instead of stating that production is challenging (via cutting propagation), it would be more beneficial to highlight studies demonstrating successful propagation through a more detailed literature review.


References
• All references should be carefully reviewed to ensure they follow the correct formatting conventions:
 Journal names should be italicized.
 Abbreviations should not be used for journal titles.
 Punctuation (e.g., "volume:") must be consistent throughout.
• Verify the completeness and accuracy of all citations.
• Attention should also be paid to the in-text citations (e.g., whether et al. should be italicized or upright, and the proper use of punctuation).

Additional comments

Corrections and suggestions have been marked in the appropriate sections of the PDF document.

Strengths:
• The study provides an optimized protocol for the micropropagation of J. procera.
• It thoroughly examines the effects of sterilization methods and growth regulators.

Weaknesses and Suggestions:
• The manuscript needs editing to improve language clarity and flow.
• The figures should be improved, and more detailed explanations should be added. Additionally, clear values can be included in the figures, or all parameters for each treatment can be expressed with their values in the text. However, this would significantly lengthen the text. Therefore, allowing readers to obtain clear values from the figures would be beneficial.
• Some statistical results should be supported with additional analyses, or the results should be presented more clearly.

---

## Round 0.2 · Minor Revisions

The reviewers have recognized a significant improvement in the manuscript, which is now clearer and the methodology is repeatable. However, some minor corrections are needed before it can be accepted. The authors are therefore asked to modify the text accordingly or, if not, to justify the choice to keep the text as it is.

Reviewer 2 ·

Basic reporting

Dear Authors,

Thank you for taking into account Reviewer’s corrections and suggestions. Now, the manuscript is clearer and easier to understand to the potential readers. Despite this, it still needs to be corrected in some parts before Acceptance.

The text is written correctly; however, the description of the results in subsection 3.3 should be revised due to inconsistencies with the cited figures. The raw data are properly presented. In my opinion, the conclusions should be corrected (suggestion in the manuscript PDF). Furthermore, some illustrations require enhancements in their description and labeling.

Experimental design

The experimental methodology is now described in detail, making it possible to repeat the experiments. Missing information on T1-T10 treatment and statistical analysis has been completed by the authors.

Validity of the findings

The presented findings are valuable for the further development of the efficient micropropagation protocols of J. procera. The performed statistical analyses of the data clearly support the stated hypothesis of the studies.
The list of references still requires detailed corrections (see attached pdf file).

Additional comments

More detailed comments and correction are included into document PDF.

Annotated reviews are not available for download in order to protect the identity of reviewers who chose to remain anonymous.

---

## Round 0.3 · Minor Revisions

Dear Authors, after a careful analysis, we found still several changes that must be made before the work can be accepted.

The “title” and the "abstract" should be re-edited by an English language expert, also the acronyms in the "abstract" (MS, BAP, etc.) should be written in full since they are cited for the first time.

Some errors in the abstract:
1. Juniper tree face → The juniper tree faces
2. The main propose → The main purpose
3. Juniper shoots are hardly to root → Juniper shoots are difficult to root
4. than → then

Pay attention to the "Conclusions" section, please read and modify where necessary. As an example, the following suggestions for modifications are reported: L. 254 "Juniper is highly depends" should say "Juniper is highly dependent"; L. 255 "proliferation rate, shoot number" should say "proliferation rate and shoot number"; L .257 "as, most " should say "as most". Please fill the "Author Contributions" section, explain what X.X. stand for. The first time you mention A.J. and A.A. please explain who you are referring to. Although error bars have been added for the other figures, they are still missing for "Figures 1 and 2". Please add such error bars, also because in the figures there are comparisons with the ANOVA test, which would be impossible without adequate data numerosity. Also specify in M&M section if and what type of transformation was done on percentage data to apply ANOVA test. Regarding the captions of the figures: in Figure 4 caption "referace" should say "refer"; in Figures 5,7, and 8 captions "T1 to T10 refer to BAP" should say "T1 to T10 refer to BAP and NAA combination” (Table, 1).

The English in the main text is much improved although there are still some errors that should be revised by an English language expert.

I leave "minor revision" decision, although the suggested corrections should be considered "mandatory".

**Language Note:** The Academic Editor has identified that the English language must be improved. PeerJ can provide language editing services - please contact us at [email protected] for pricing (be sure to provide your manuscript number and title). Alternatively, you should make your own arrangements to improve the language quality and provide details in your response letter. – PeerJ Staff

---

## Round 0.4 · accepted · Accept

The authors have addressed the comments and the manuscript is ready for publication.